# mTOR Complex 1 Content and Regulation Is Adapted to Animal Longevity

**DOI:** 10.3390/ijms23158747

**Published:** 2022-08-06

**Authors:** Natalia Mota-Martorell, Mariona Jové, Reinald Pamplona

**Affiliations:** Department of Experimental Medicine, University of Lleida, Lleida Biomedical Research Institute (UdL-IRBLleida), E-25198 Lleida, Spain

**Keywords:** mTORC1, longevity, ageing, age-related diseases, metabolism

## Abstract

Decreased content and activity of the mechanistic target of rapamycin (mTOR) signalling pathway, as well as the mTOR complex 1 (mTORC1) itself, are key traits for animal species and human longevity. Since mTORC1 acts as a master regulator of intracellular metabolism, it is responsible, at least in part, for the longevous phenotype. Conversely, increased content and activity of mTOR signalling and mTORC1 are hallmarks of ageing. Additionally, constitutive and aberrant activity of mTORC1 is also found in age-related diseases such as Alzheimer’s disease (AD) and cancer. The downstream processes regulated through this network are diverse, and depend upon nutrient availability. Hence, multiple nutritional strategies capable of regulating mTORC1 activity and, consequently, delaying the ageing process and the development of age-related diseases, are under continuous study. Among these, the restriction of calories is still the most studied and robust intervention capable of downregulating mTOR signalling and feasible for application in the human population.

## 1. Introduction

The mechanistic (or mammalian) target of the rapamycin (mTOR) signalling pathway is one of the most studied intracellular networks, regulating the ageing and longevity processes. Its description is closely related to the discovery of the antifungal rapamycin, which was isolated from *Streptomyces hygroscopius* in the late 1970s. Immunosuppressant and antitumor activities were attributed to rapamycin before its mechanism of action was understood. In the 1990s, the genes codifying the putative targets of rapamycin in yeast (*target of rapamycin (TOR) 1* and *TOR2*) were identified, but it wasn’t until 1994 that the mammalian homolog was isolated from mammalian cells [1].

mTOR is a member of an evolutionarily conserved group of serine/threonine kinases from the phosphatidyl-inositol-3 kinases (PI3K)-related kinase family that is present as two distinct complexes: mTOR complex 1 (mTORC1) and mTOR complex 2 (mTORC2) [2]. It has been suggested that mTORC1 and 2 regulate different processes. In fact, mTORC1 is sensitive to rapamycin, and its inhibition has been widely associated with an extended longevity and age-related processes. Therefore, most scientific efforts have been directed towards studying this complex. However, in recent years, mTORC2 has also been an aim of study, and its integrity has recently been suggested to regulate longevity in worms [3].

mTORC1 plays a central role in the highly conserved mTOR signalling network, which integrates intracellular and extracellular signals to regulate cell physiology and metabolism [4]. The downstream effectors of mTOR regulate multiple processes, including mRNA translation, protein and lipid biosynthesis, mitochondrial function and biogenesis, stress responses, and autophagy, among others [2,4,5,6,7,8,9]. Briefly, under favourable conditions for cell growth, mTORC1 is activated and promotes anabolic processes. Conversely, when nutrient availability is scarce, mTOR is not activated, and catabolic processes are induced in order to feed the cells and maintain proper energy levels.

## 2. The Structure and Regulation of the mTORC1

Biochemical studies and cryo-electron microscopy allowed the description of the mTORC1 structure, which is defined as a giant complex weighing 1 MDa [10]. In mammals, the core of mTORC1 forms a symmetric dimer of an heterotrimer consisting of the mTOR itself, Raptor (regulatory associated protein of TOR) and mLST8 (mammalian lethal with SEC13 protein 8) [2] (Figure 1). These core elements of the mTORC1 are associated with inhibitory subunits, such as Deptor (DEP domain-containing mTOR-interacting protein) and PRAS40 (Proline-rich AKT1 substrate of 40 kDa). Deptor physically interacts with mTOR, inhibiting either mTORC1 or mTORC2 [11]. PRAS40 is a polypeptide that binds to Raptor, and thus it is commonly stably associated with mTORC1, which is repressed [10]. However, under certain conditions, PRAS40 is phosphorylated and separated from mTORC1, which is released from its inhibitory regulation. FKBP12 (FK506 binding protein) is a regulatory subunit of the rapamycin-sensitive mTORC1 activity. Importantly, Raptor and mTOR regulators PRAS40 and FKBP12 are exclusive to mTORC1.

The regulation of mTORC1 is complex, and occurs at two levels: their upstream effectors and within mTORC1 itself. The upstream regulation of mTORC1 depends upon both extra- and intracellular signals. Specifically, extracellular signals include nutrient availability (e.g., glucose or trophic factors such as insulin), amino acids (e.g., leucine and arginine), and oxygen, whereas high energy demands (low ATP/ADP ratio) and metabolism intermediates (e.g., methionine and related metabolites) are sensed inside the cells [2,14,15]. These signals are transduced through multiple proteins and protein complexes, which are regulated through different phosphorylation patterns, finally activating mTORC1.

Notably, within mTORC1 there are also specific modifications that are characteristic of active mTORC1. In fact, multiple mTOR phosphorylation sites have been described, with the activating phosphorylation of Ser^2448^ and Thr^2446^ via Akt being the most commonly used. However, mTOR phosphorylation at Ser^2448^ and Thr^2446^ via S6K1 and AMPK can be inhibitory [16]. In fact, mTORC1 mutations at Ser^2448^ do not prevent the phosphorylation of its downstream effector S6K1, and deletion of Ser^2448^ surrounding amino acids lead to mTORC1 hyperactivation [17], revealing the complexity of phosphorylation in mTORC1 activity regulation. Other phosphorylation sites of mTOR are reviewed in [16]. Fewer phosphorylation sites have been described for PRAS40, which can be phosphorylated at Thr^246^ via AKT [18], but also at Ser^183^ and Ser^221^ via mTOR [19]. The phosphorylation of PRAS40, either via AKT or mTOR, leads to its dissociation from mTORC1, which is released and active [20]. Given this complexity, multiple phosphorylation patterns regulate mTORC1. For instance, phosphorylated mTOR at Ser^2448^ and dephosphorylated PRAS40 (at Thr^246^) results in an active mTORC1, whereas dephosphorylated mTOR (at Ser^2448^) and phosphorylated PRAS40 (at Thr^246^) lead to an inactive mTORC1 [14].

Finally, active mTOR promotes the induction of anabolic processes, such as lipid and protein biosynthesis, for maintaining proper cellular structure and function and mitochondrial biogenesis and metabolism, in order to fulfil the energetic requirement to maintain biosynthetic processes. Conversely, when cells are exposed to scant nutrient availability or hypoxic conditions, mTOR is not activated, and catabolic processes such as autophagy and the inhibition of biosynthetic processes are promoted in order to feed the cell and maintain proper energetic levels. Recent evidence points to the existence of a regulatory loop through which autophagy, after the recycling of cellular components, leads to the reactivation of mTORC1 [21].

## 3. mTORC1 and Longevity

Reduced mTOR signalling has been widely associated with longevity [2] (Table 1). In fact, apart from having a role in determining species longevity, the mTOR signalling pathway seems to be a key determinant of individual longevity [5]. Accordingly, reduced activity of the mTOR signalling network has been reported in tissues from exceptionally long-lived rodents [22,23], whales [24], and other mammals [25].

Molecular adaptations in the mTOR signalling pathway leading to an enhanced longevity seem not only to be restricted to the global mTOR signalling network, but also to mTORC1 itself (Figure 2). In fact, the existence of a longevity-associated mTORC1 genomic, proteomic and metabolomics profile in the hearts of mammals has been recently described [14], and is characterised by reduced gene expression and protein content of mTORC1 subunits and activators. Specifically, the authors reported a reduced content of mTOR and Raptor, and identified phosphorylation patterns in mTOR and PRAS40 that might be consistent with a reduced mTORC1 activity in long-lived species. These results are supported by recently published transcriptomics data [25], and extend to other mammalian tissues. Again, the longevity-associated phenotype was associated with reduced mTORC1 core elements, such as *mtor* and *rptor*; downstream effectors such as *rps6kb1* (S6K1); and its negative regulators *akt1s1* (PRAS40) and *fkbp1a* (FKBP12). Accordingly, reduced mRNA content of *rptor* (Raptor) and mTORC1-negative regulators, such as *akt1s1* (PRAS40), was found in blood from nonagenarians [26]. In addition, the offspring of those long-lived individuals have reduced *rptor* gene expression, which has emerged as a potential biomarker of familial longevity [26]. Altogether, these results suggest that not only protein content and gene expression mTORC1 core elements, but also that its regulatory factors are reduced in long-lived species and individuals.

Experimental regulation of mTOR signalling or mTORC1 content itself, through the generation of genetically modified animal models, affects longevity (Table 2). In fact, the genetic inhibition of upstream regulators [27,28], downstream effectors [29,30], or mTORC1 components themselves [27,31,32,33,34] leads to a reduced mTORC1 content and increased longevity in invertebrate [27,31,32] and mice models [28,29,30,33,34]. In addition, in rodents, reduced mTORC1 content leads to an improved health span [28,29,30] and delayed ageing [34]. In contrast, mTOR activation shortens longevity [1,6,8] and leads to altered behaviour and neuronal function in rodents [35]. Taken together, these studies support the idea that reduced mTORC1 content and activity, as well as its integrity, play central roles in longevity determination and enhanced health span.

The evolution of longevity occurred through mTOR regulation. These adaptations evolved by maintaining lowered steady-state levels of mTORC1 content or activity [14,25,26,33,34], reducing the content of mTORC1 upstream regulators [14,25,26,28], or lowering the activation of its downstream effectors [14,22,23,25,29,30]. Furthermore, these regulations operate at all expression levels, either genes [14,25,26], proteins [14,22,23], or metabolites [14]. Apart from the metabolic effects resulting from a reduced pathway activity, reduced mTORC1 is essential to maintaining its integrity [35].

## 4. mTORC1 and Ageing and the Development of Age-Related Diseases

mTOR content [36] and activity [37,38,39] are increased in aged rodents (Table 3). These changes are concomitant with altered content of its downstream effectors. In fact, increased S6K1 [38] and PGC1α [36], along with reduced Atg13 [36], have been found in old rodents compared to young. Taken together, these results support the idea that, in aged specimens, there is an alteration in mitochondrial biogenesis, lipid and protein metabolism, oxidative stress, and autophagy, which are intracellular processes regulated through mTOR. Accordingly, increased electron leakage and ROS, lipoxidation and nuclear DNA alteration, and protein ubiquitination [36,38] are intracellular age-associated traits in rodents. These results are supported by Tang and collaborators, who performed transcriptomic profiling of muscle fibres from hyperactivated mTOR (TSC^−/−^) aged mice, revealing that mTORC1 activation increased the expression of genes involved in oxidative stress and catabolic processes [40].

The age-dependent increase in mTORC1 activity is not continuous and constant throughout the lifespan. In fact, in skeletal muscle from rats, mTOR phosphorylation is sustained in young and adult rodents, demonstrating a marked increase in middle-aged specimens [38]. Interestingly, S6K1 activity and protein ubiquitination show a pattern of increase in middle age that is similar to that of mTOR, while Akt phosphorylation upstream of mTORC1 and 4EBP1 phosphorylation downstream of mTORC1 increase continuously with age. Similarly, in frontal cortex from healthy humans aged between 40 and 79 years, a marked and transitory increase in mTOR content and phosphorylation from samples aged in their 60 s was described [41]. In the same study, the authors described a continuous accumulation of lipoxidation and protein oxidation markers, which was enhanced from the age of 65 years. Later, the authors obtained similar results regarding fatty acid profile [42]. In fact, they described that fatty acid composition is maintained through the lifespan, until an age of 65 years. At this point, there is an increase in the content of saturated fatty acid, along with a reduced content of unsaturated fatty acids, mostly those with high degrees of unsaturation. This is interesting, because results from these studies [38,41,42] suggest that a transitory induction of mTOR activity in the frontal cortex might lead to a cascade of events, mainly regulating lipid and protein biosynthesis, that are exacerbated during the ageing process.

Most of the mentioned processes are altered during ageing, and lead to the development of age-related diseases. In fact, oxidative stress and altered protein metabolism leading to insoluble aggregates and membrane unsaturation are processes under the control of mTOR that have been associated with the development of Alzheimer’s disease (AD) [43,44]. mTORC1 activation has been associated with neuronal death in AD, making it a potential therapeutic target [43,45]. Along the same lines, treatment of SH-SY5Y cells with amyloid β peptide has been reported to lead to increased expression of mTOR and reduced expression of its inhibitor, Deptor [46]. Furthermore, the protein content of Deptor is increased in the gyrus cingulate and the occipital lobe in AD patients compared to control, and reduced in samples from late-onset AD compared to early-onset familial AD [46]. Deletion of FKBP12 in mice leads to increased content and phosphorylation of mTOR in the hippocampus, along with altered behaviour and neuronal function [35]. Interaction between mTOR and the scaffold Raptor is enhanced, ensuring mTORC1 integrity and suggesting that not only mTOR content and function, but also maintenance of its integrity, play a role in brain ageing.

In addition to their potentially beneficial neurological effects, reduced protein biosynthesis and enhanced protein degradation might lead to reduced muscle mass and sarcopenia. Paradoxically, muscle mass decay in aged specimens is concomitant with mTORC1 activation in skeletal muscle. Furthermore, it has been demonstrated that inhibition of mTORC1 with rapamycin prevents almost all deleterious processes associated with skeletal muscle ageing [47,48], and its genetic activation is sufficient to promote molecular mechanisms associated with sarcopenia [47].

## 5. mTORC1 and Nutritional Interventions in Ageing

The effect of mTORC1 inhibitors in ageing has been a matter of scientific study since the 1990s. In fact, most initial efforts elucidating the beneficial effects of rapamycin have revealed its role in ageing, with Romero et al. describing in 1995 its effect in preventing reduced bone growth in aged rat [49]. Since mTORC1 responds to nutritional availability, the effects of mTORC1 inhibition using specific inhibitors such as rapamycin or through nutritional interventions in ageing and health have been widely reviewed.

### 5.1. Rapamycin Treatment

Short-term rapamycin treatment is enough to reduce mTOR content and signalling in multiple tissues, including heart, kidney, liver, intestine, and visceral fat [36,50,51] and signalling [52,53]. Remarkably, when injected intraperitoneally, reduced organismal mTOR signalling has been reported after only 1 h [53]. The reduced mTOR activity occurs along with an enhanced longevity [36,50,51,53,54,55,56]. Interestingly, rapamycin reduces mTOR activity and increases longevity in a dose-dependent manner [50] (Table 4).

In old rodents, short-term rapamycin intake is enough to significantly increase longevity [57,58] and reduce mortality [38]. The health span is also improved, as it is associated with delayed ageing [36,58], increased global activity [56], reduced tumour incidence [56], and enhanced immunity [37]. Additionally, decline in muscular function and motor coordination [58] and hematopoietic stem cell function [37] with age are preserved.

The molecular mechanisms through which rapamycin extends longevity and delays ageing are diverse, and include reduced ROS production at complex I, protein and lipid oxidation, and mitochondrial DNA alteration, along with enhanced mitochondrial biogenesis, stress resistance, autophagy and modulation of lipid metabolism [36,50,51,54,55]. The effects on protein translation and chromatin structure are complex. Intuitively, mTORC1 inhibition should lead to a translation repression [59], as has been reported [50]. However, a recent study revealed that rapamycin induces histone expression in the intestine of female flies [51]. Accordingly, previous studies have already described the existence of non-canonical mechanisms of protein synthesis mediated though eIF3, an mTOR downstream effector, which remains to be completely elucidated [60,61]. Hence, these data point to a complex mechanism through which mTORC1 could lead to the synthesis of specific proteins such as histones, altering chromatin and repressing global translation, thus saving energy under starving conditions.

Interestingly, most of the rapamycin-mediated effects on health and longevity are sex dependent. In fact, mTOR signalling is globally depleted in females after treatment, whereas it was only reduced in males [53]. Accordingly, most studies reported that the lifespan extension effect is higher in female compared to male [50,56], even when applied at old ages [52,57].

Females also seem to be more sensitive to the adverse effects of rapamycin, which are dependent on administration patterns, specifically high dosage and long periods. Accordingly, dosage of rapamycin (8 mg/kg intraperitoneally [58] or 1.5 mg/kg subcutaneously [62]) is associated with higher incidence of chromosome aberration in bone marrow cells, and higher cancer incidence and invasiveness. The effects of a high dosage of rapamycin are unclear, as short-term treatment did not alter longevity [58], although high-dose administration throughout the lifetime, starting at young age, extended longevity [62]. Moderate rapamycin treatment for 1000 d is associated with a higher incidence of cancer (64% as a cause of death) compared to controls (46%) [52]. Interestingly, and probably associated with the previously mentioned chromosome aberration [58], the most prevalent carcinoma was lymphoma [56,62].

These protocols might lead to altered plasma rapamycin levels. In fact, although not significant, two studies have reported that slightly lower rapamycin plasma levels is associated with higher longevity [52,58], even when mTOR signalling is significantly lowered [52], and independently of sex. These results suggest that it might be worth deeply studying rapamycin metabolism and the association between its plasma levels and its effects on health. Nonetheless, more studies are needed to elucidate the appropriate administration pattern and dosage, as has been described previously [63,64].

**Table 4 ijms-23-08747-t004:** mTOR regulation through rapamycin. Increased or reduced mTOR content can refer to either transcript content gene expression, protein content, protein phosphorylation or activity after insulin stimulation. Letters refer to: maximum longevity (ML), in years; female (F); male (M); not determined (n.d.). Nutritional intervention (NI) duration is grouped according to different time periods: ^1^ very short-term (hours to days); ^2^ short-term (1 to 6 months); ^3^ long-term (more than 6 months); ^4^ lifetime (to natural death of the specimen). The beginning of the NI on is defined by superscript letters: ^A^ NI applied to young specimens; ^B^ NI applied to adult specimens; ^C^ NI applied to middle-aged specimens; ^D^ NI applied to old specimens. Symbols refer to: * Studies in which the expression, content or phosphorylation of mTORC1 itself were evaluated; ^#^ Studies in which mTORC1 itself was not evaluated, but its upstream or downstream effectors were.

Species	Sex	Intervention	mTORC1	Tissue	Phenotype	Longevity	Ref.
Worm	-	Rapamycin ^A^	Reduced *	Whole	n.d.	+21%	[55]
Fly	M/F	Rapamycin ^D^	Reduced ^#^	Whole, abdomen, thorax and head	n.d.	+18%	[50]
Fly	F	Rapamycin ^D^	Reduced ^#^	Intestine	n.d.	+8%	[51]
Fly	M/F	Rapamycin ^D^	Reduced	n.d.	n.d.	+33%	[54]
Mouse	M/F	Rapamycin ^A2^	Reduced ^#^	Heart, liver, kidney and intestine	n.d.	n.d.	[53]
Mouse	M	Rapamycin ^B3^	Reduced ^#^	Liver	Delayed ageing	n.d.	[36]
Mouse	F	Rapamycin ^B4^	Reduced	n.d.	n.d.	+4%	[57]
Mouse	M	Rapamycin ^B4^	Reduced	n.d.	n.d.	+5%	[57]
Mouse	F	Rapamycin ^B4^	Reduced	n.d.	Delayed ageing	+9–14%	[58]
Mouse	M	Rapamycin ^B4^	Reduced	n.d.	Delayed ageing	+14%	[58]
Mouse	M	Rapamycin ^B4^	Reduced *	n.d.	n.d.	+60%	[37]
Mouse	F	Rapamycin ^D1^	Reduced	n.d.	Reduced cancer	+9%	[62]
Mouse	F	Rapamycin ^D2^	Reduced	n.d.	n.d.	+9–15%	[56]
Mouse	M	Rapamycin ^D2^	Reduced	n.d.	n.d.	+15–18%	[56]
Mouse	F	Rapamycin ^C2/D4^	Reduced ^#^	Visceral fat	n.d.	+14%	[52]
Mouse	M	Rapamycin ^C2/D4^	Reduced ^#^	Visceral fat	n.d.	+9%	[52]
Mouse	F	Rapamycin ^D4^	Reduced	n.d.	n.d.	+12%	[57]
Mouse	M	Rapamycin ^D4^	Reduced	n.d.	n.d.	+9%	[57]

### 5.2. Calorie and Amino Acid Restriction

Calorie restriction (CR) is defined as a reduced calorie intake without inducing malnutrition. It is widely accepted that CR is the only nutritional intervention capable of robustly extending longevity in animal models and promoting health span. Additionally, it has been elucidated that most of these effects are due to the specific restriction of calories from protein intake [65], rather than those from lipids [66] or carbohydrates [67]. Accordingly, isocaloric protein restriction (PR) is responsible for most of these effects [68]. Among all the amino acids constituting proteins in living beings, it has been demonstrated that the selected isocaloric restriction of sulphur containing the amino acid methionine (MetR) accounts for most, if not all, of the beneficial effects attributed to CR and PR [68].

Not surprisingly, CR, PR and MetR prevent mTOR signalling activation, by either regulating mTOR content [69,70,71] and its phosphorylation [38,69,70,71,72,73], its downstream effector S6K [38,72,73,74], or its upstream regulator TSC [73] (Table 5). The regulation of mTOR signalling is complex, since while mTOR phosphorylation is dose dependent below 5% to 40%CR regimens [73], S6K and TSC regulation is dual. In fact, 5%CR is enough to reduce S6K phosphorylation, although more intense CR (40%) is needed to reduce TSC phosphorylation. In old specimens, 40%CR [38] or 80%MetR [75] are sufficient to lower mTOR signalling and to restore it to the basal levels of activation found in young specimens, delaying ageing [38].

CR, PR and MetR promote health span at organismal levels, including reduced body weight and percentage of fat mass, lower blood glucose and insulin, and lower plasma TG and cholesterol [72,75,76,77], suggesting improved metabolic health. Learning and memory processes are also improved [70,71,77,78,79], and tumour incidence is reduced [80]. At the molecular level, the intracellular processes regulated by mTOR upon restrictions include increased ROS and NRF2 target genes to maintain proper hormesis, increased expression of genes related to fatty acid oxidation peroxisome β-oxidation, as well as pro-apoptotic genes and reduced neuroinflammation [71,74,76,78].

Ageing leads to sarcopenia, which is regulated though mTORC1, as described previously. Among all the organismal effects, 40%CR in skeletal muscle from middle aged specimens led to mTORC1 inhibition and reduced protein degradation, contributing to limiting age-related sarcopenia [74]. However, DR [76] and PR [74] also led to reduced plasma and hepatic content of Met, which is essential for starting protein synthesis in eukaryotes and could affect muscle maintenance during ageing. Recently, it was described that the initial sensing of MetR occurs in the liver, where a stress response for reducing Met use and protein synthesis is activated [81]. These molecular processes depend upon mTOR, and are directed to restore basal Met levels after 4 days, hence making it possible to maintain proper protein synthesis.

The production of ROS, and the accumulation of oxidized biomolecules constitutes one of the main causes of ageing. Recently, it has been reported that CR and MetR benefits, in terms of oxidative damage, depend on sulphur disulphide (H_2_S) production [75,76]. This molecule is a by-product of the transulphuration pathway (TSP), which relates two sulphur-containing amino acids, such as Met and cysteine. In rodents undergoing CR and MetR, tissue TSP intermediates are reduced, along with an enhanced content of TSP-related enzymes and H_2_S levels [75,76]. Consequently, tissue damage is lowered. Additionally, when mTORC1 is constitutively active, H_2_S production is depleted, and Met levels are reduced but maintained at a higher level than that of the control [76]. Evidence also points to H_2_S as a neuroprotective factor for preventing the development of neurodegenerative diseases [78]. Taken together, these data make it possible to establish a relationship between CR, Met and mTOR signalling. Supporting this data, Gu and collaborators described the existence of SAMTOR, a protein that inhibits mTOR after sensing nutrient availability via SAM, a metabolite derived from methionine [82].

Branched-chain amino acids (BCAA) are amino acids with an aliphatic side-chain, and include leucine (Leu), isoleucine (Ile) and valine (Val). Recently, the restriction of BCAA (BCAAR) has been found to be associated with improved health span and longevity, although most of these effects are sex dependent [39]. Although BCAAR improves metabolic health in both young and old male and female mice, most of the beneficial effects are limited to males. Accordingly, lifespan is increased and frailty is reduced in aged specimens, along with a reduction in mTOR signalling. Conversely, in females, mTOR signalling is unaffected, and it is associated with higher early mortality. Accordingly, BCAA supplementation shortens lifespan and worsens metabolic health, although these processes seem to be independent of mTOR [83]. In humans, supplementation with BCAA is neurotoxic, and promote oxidative stress and inflammation in peripheral blood mononuclear cells through mTORC1 activation [84].

Although the mechanisms through BCAA extend their effects, Leu depletion leads to improved metabolic health and reduced mTOR phosphorylation and signalling [72,85]. Restriction of Leu (LeuR) and MetR act through similar pathways, since some of the beneficial effects induced by MetR can also be observed under LeuR [72]. However, MetR emerges as the most robust intervention, as it induces more intense effects.

**Table 5 ijms-23-08747-t005:** mTOR regulation through nutritional interventions. Increased or reduced mTOR content can refer to transcript content, gene expression, protein content, protein phosphorylation or activity after insulin stimulation. Letters refer to: maximum longevity (ML), in years; dietary restriction (DR); protein restriction (PR); branched-chain amino acids restriction (BCAAR); leucine restriction (LeuR); methionine restriction (MetR); female (f); male (m); not determined (n.d.). Nutritional intervention (NI) duration is grouped according to different time periods: ^1^ very short-term (hours to days); ^2^ short-term (1 to 6 months); ^3^ long-term (more than 6 months); ^4^ lifetime (to natural death of the specimen). Beginning of the NI on is defined by superscript letters: ^A^ NI applied to young specimens; ^B^ NI applied to adult specimens; ^C^ NI applied to middle-aged specimens; ^D^ NI applied to old specimens. Symbols refer to: * Studies in which the expression, content or phosphorylation of mTORC1 itself were evaluated; ^#^ Studies in which mTORC1 itself wasn’t evaluated, but its upstream or downstream effectors were.

Species	Sex	Intensity	mTORC1	Tissue	Phenotype	Longevity	Ref.
Mouse	M/F	CR (50%) ^A1^	Reduced	-	n.d.	n.d.	[76]
Rat	F	CR (40%) ^B1^	Reduced *	Liver	n.d.	n.d.	[73]
Human	F	CR (−1000 kcal) ^B3^	Reduced	Amygdala	Improved function and metabolic health	n.d.	[77]
Rat	F	CR (40%) ^B3^	Reduced *	Skeletal muscle	Delayed ageing	n.d.	[38]
Humans	F/M	CR (30%) ^B4^	Reduced	-	Improved memory and metabolic health	n.d.	[79]
Mouse	M	CR (30%) ^C1^	Reduced *	Hippocampus (neurons)	n.d.	n.d.	[69]
Mouse	M	CR (30%) ^C1^	Reduced *	Hippocampus (neurons)	Improved learning and memory	n.d.	[70]
Mouse	M	CR (30%) ^C1^	Reduced *	Hippocampus (astrocytes)	n.d.	n.d.	[71]
Mouse	F	PR (100%) ^A1^	Reduced ^#^	Liver	n.d.	n.d.	[74]
Mouse	F	PR (7–21%) ^D1^	Reduced ^#^	Liver, heart skeletal muscle, adipose tissue	n.d.	n.d.	[80]
Mouse	F	MetR (80%) ^B1^	Reduced	Hippocampus, cortex	Improved learning and memory	n.d.	[78]
Mouse	M	MetR (80%) ^B2^	Reduced *	Liver	Improved metabolic health	n.d.	[72]
Mouse	M	MetR (80%) ^B4^	Reduced *	Kidney	Delayed ageing	n.d.	[75]
Mouse	M/F	MetR (80%) ^D1^	Reduced *	n.d.	n.d.	+25%	[75]
Mouse	M	BCAAR (70%) ^C1/C3/D1^	Reduced	Heart, liver	Delayed ageing	+12.3	[39]
Mouse	F	BCAAR (70%) ^D1/D3^	Reduced	n.d.	Improved metabolic health	n.d.	[39]
Mouse	M	LeuR (80%) ^B2^	Reduced *	Liver	n.d.	n.d.	[72]
Mouse	M	LeuR (0%) ^B2^	Reduced *	Liver	n.d.	n.d.	[85]

## 6. mTORC1 and Nutritional Interventions in Pathological Processes

Aberrant hyperactivated mTORC1 has been associated with the development of pathological processes, such as AD (Table 6). Accordingly, in AD pathogeny, mTOR is aberrantly hyperphosphorylated and hyperactivated [43]. In addition, the degree of mTOR phosphorylation is correlated with AD progression. Consequently, autophagy function is altered, leading to the accumulation of protein aggregates such as amyloid plaques, which are constituted of neurofibrillary tangles and hyperphosphorylated tau, which are molecular hallmarks of AD [86]. Since autophagy is regulated through mTOR activation, rapamycin has emerged as a potential drug for treating AD [86]. Accordingly, in mouse models of AD carrying human APP, rapamycin is able to reduce amyloid plaque, as well as to prevent neurovascular coupling and memory impairment [45].

Multiple lines of evidence also point to CR as a feasible strategy not only for treating, but also preventing the development of neurodegenerative disorders. Regarding AD, in healthy specimens, 30%CR is enough to reduce the number and size of age-associated accumulation of amyloid plaques, astrocyte activation and Tau phosphorylation in multiple brain regions, including the hippocampus, cortex, neocortex, dentate gyrus, and entorhinal cortex [87,88,89,90]. These effects are concomitant with improved locomotion and memory processes. Again, Met might play an important role in driving CR-neuronal beneficial effects. Evidence suggests that Met promotes neuroinflammation and impairs neurogenesis [91]. Met is also associated with memory loss [92], due to the alteration of methylation patterns, and thus the expression of Netrin-1, which is involved in neuronal processes. Since Netrin-1 inhibits mTORC1 [93], these results suggest that memory impairment in AD might be associated with hyperactivation of mTORC. However, no studies evaluating the effect of MetR in AD animal models have yet been performed.

The role of the mTOR pathway in the development of cancerous processes has also been a focus of study [94] (Table 6). In fact, tumours are also characterized by the hyperactivation of the mTOR signalling pathway, leading to increased cell proliferation and invasiveness [95]. Accordingly, rapamycin [80,95], as well as CR and PR [73,80,96], is capable of reducing mTOR activation in tumour cells and reducing cancer incidence and tumour size, probably due to an enhancement of autophagy and a reduction in protein synthesis. The underlying processes that lead to this phenotype are not well understood, although they might be associated with the fact that mTOR acts as a metabolic checkpoint, and its inhibition leads to S-phase arrest in KRas cells due to glutamine deprivation [97]. mTOR is also responsible for mediating T cell differentiation through metabolic adaptations, and for promoting immune response to cancer processes [98]. Notably, rapamycin is not, per se, used for human cancer treatment, due to its poor solubility and pharmacokinetics. Instead, paralogs of rapamycin are undergoing clinical trials for the treatment of several cancers [94].

This is supported by the role of mTORC inhibition in delaying ageing, and subsequently, preventing the development of age-related disease in mouse models of progeria and accelerated ageing, revealing that genetic disruption of mTOR, as well as BCAAR, leads to improved health and delayed ageing [99].

**Table 6 ijms-23-08747-t006:** mTOR regulation through nutritional interventions in pathological processes. Increased or reduced mTOR content can refer to transcript content, gene expression, protein content, protein phosphorylation or activity after insulin stimulation. Letters refer to: maximum longevity (ML), in years; dietary restriction (DR); protein restriction (PR); branched-chain amino acids restriction (BCAAR); female (f); male (m); not determined (n.d.). Nutritional intervention (NI) duration is grouped according to different time periods: ^1^ very short-term (hours to days); ^2^ short-term (1 to 6 months); ^3^ long-term (more than 6 months); ^4^ lifetime (to natural death of the specimen). The commencement of NI is defined by superscript letters: ^A^ NI applied to young specimens; ^B^ NI applied to adult specimens; ^C^ NI applied to middle-aged specimens; ^D^ NI applied to old specimens. Symbols refer to: * Studies in which the expression, content or phosphorylation of mTORC1 itself were evaluated; ^#^ Studies in which mTORC1 itself was not evaluated, but its upstream or downstream effectors were.

Species	Sex	Disease	Intervention	mTORC1	Tissue	Phenotype	Ref.
Mouse	M	AD	Rapamycin ^B1^	Reduced	Cortex	Improved health	[45]
Mouse	M	AD	Rapamycin ^B2^	Reduced	Brain	Improved health	[45]
Mouse	M	AD	Rapamycin ^C1^	Reduced	Brain	Improved health	[45]
Mouse	M	AD	CR (40%) ^B1^	Reduced	Hippocampus, cortex miand dentate gyrus	Improved health	[87]
Mouse	MF	AD	CR (30%) ^B1^	Reduced	Hippocampus	Improved health	[88]
Mouse	F	AD	CR (30%) ^C1^	Reduced	Hippocampus, neocortex	Improved health	[90]
Mouse	F	AD	CR (30%) ^C1^	Reduced	Hippocampus, cortex, entorhinal cortex	Improved health	[89]
Mouse	F	Cancer	Rapamycin ^B1^	Reduced	Tumour	Reduced tumour size	[95]
Mouse	F	Cancer	Rapamycin ^B1^	Reduced	n.d.	Reduced cancer incidence	[100]
Mouse	F	Cancer	DR (50%) ^A1^	Reduced	Tumour	Reduced tumour size	[96]
Rat	F	Cancer	DR (40%) ^B1^	Reduced *	Tumour	Reduced cancer incidence	[73]
Mouse	F	Cancer	PR (7–21%) ^D1^	Reduced ^#^	Tumour	Reduced tumour size	[80]
Rat	M	Diabetes	PR (25%) ^A2^	Reduced ^#^	Kidney	Improved health	[101]
Mouse	M/F	Progeria	mTOR ^(Δ/+)^	Reduced *	Fibroblasts	Improved health	[99]
Mouse	M/F	Progeria	BCAAR ^D2^	Reduced	n.d.	Improved health	[39]

## 7. Clinical Trials Aimed at Delaying the Ageing Process

Rapamycin, as well as its analogue, Sirolimus, was approved by the FDA in 1999 for use as an immunosuppressant. Accordingly, its role in the treatment of cancer has been widely reviewed [102,103], and multiple clinical trials evaluating its efficacy for the treatment of multiple cancers and the development of neurological diseases are ongoing.

Conversely, despite knowledge regarding the molecular mechanisms linking mTOR modulation and longevity and ageing, little is known regarding the effect of mTOR inhibition in the prevention of the ageing process in humans. In fact, only four studies have previously evaluated the effect of mTOR inhibition in healthy individuals. Among these, only one was performed in aged individuals (aged between 70 and 95 years), and it revealed that short-term administration of rapamycin (oral, 8 weeks), was not enough to improve cognitive function, physical performance, or individual self-perception of health status [104]. Two studies revealed that oral administration of a unique mTOR inhibitor (RTB101 [105]) or a combination (catalytic inhibitor BEZ235 plus RAD001, an allosteric and selective inhibitor of mTORC1 [106]) improved immune system function.

Along the same lines, mTORC1, but also mTORC2, regulates lineage specification in the immune system. It drives proinflammatory T-cell expansion, but also anti-inflammatory macrophage polarization [107]. Given its role in inflammatory processes, mTOR inhibitors are being developed to treat autoimmune degenerative diseases, such as systemic lupus erythematosus (SLE) [107], but also obesity, and to improve cardiovascular health [108]. In patients with SLE, treatment with rapamycin promotes the development of regulatory CD4 T cells and memory T cells, which are essential for reducing inflammation in SLE [109]. In these regulatory SLE T cells, inhibition of mTOR restores autophagy and its suppressor function [110]. Since patients with rheumatic diseases have a shortened lifespan, regulation of inflammatory processes via mTOR inhibition appears to be a critical mechanism of lifespan extension in humans. In addition, topical administration of rapamycin also prevented skin ageing [111].

Taken together, these results suggest that the use of selective inhibitors of mTOR is a promising and emerging mechanism for preventing ageing and the development of associated comorbidities. However, more studies are needed to evaluate its effects in healthy individuals. Actually, the Participatory Evaluation (of) Aging (with) Rapamycin (for) Longevity Study (PEARL, NCT04488601), is the only ongoing clinical trial aimed at evaluating the effect of rapamycin in healthy individuals aged between 50 and 85 years.

## 8. Conclusions

The signalling mTOR pathway, and mTORC1 itself, are key regulators in the determination of longevity and ageing. In fact, its inhibition is associated with enhanced longevity, whereas its activation leads to ageing and the development of age-associated diseases. Since mTOR responds to nutrient availability and regulates overall intracellular metabolism, it becomes a target to be modulated in order not only to treat but to prevent the development of age-related diseases. Accordingly, evidence points to rapamycin as a feasible candidate for modulating ageing and treating AD and cancer. However, the optimal administration pattern for enhancing and promoting beneficial effects remains to be elucidated. Overall, CR appears to be the most consistent nutritional intervention for inhibiting mTOR—and thus delaying ageing and preventing the development of comorbidities associated with age—that is feasible for application in the human population.

## Figures and Tables

**Figure 1 ijms-23-08747-f001:**
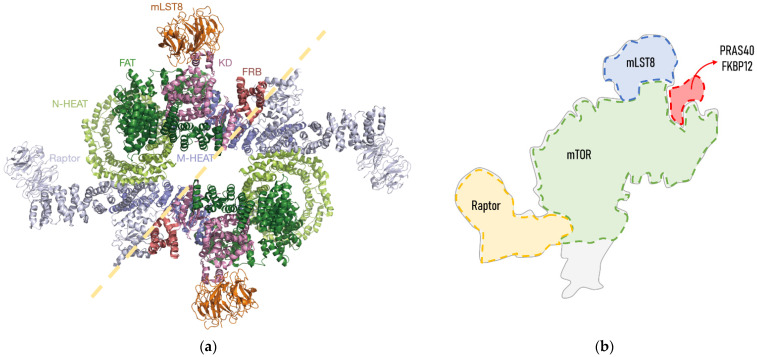
Structure of human mTORC1: (**a**) Cryo-electron microscopy structure obtained from Yang et al., 2018. mTORC1 forms a symmetric dimmer of an heterotrimer consisting of mTOR itself, Raptor and mLST8. The mTOR structure consists of N-terminal HEAT repeats (N-HEAT), an FATD (Frap, ATM, TRRAP domain), an FRBD (FKBP12-rapamycin binding domain) and a C-terminal kinase domain (KD). Each of the domains and subunits are indicated in different colours; (**b**) schematic representation of an mTORC1 heterotrimer and expected binding site of its regulators FKBP12 and PRAS40, adapted from [12,13]. FKBP12 binds to mTOR through its FRBD. The PRAS40 binding domain is structurally very close to that of FKBP12, as it binds to an α-helix from the FBD and to mLST80.

**Figure 2 ijms-23-08747-f002:**
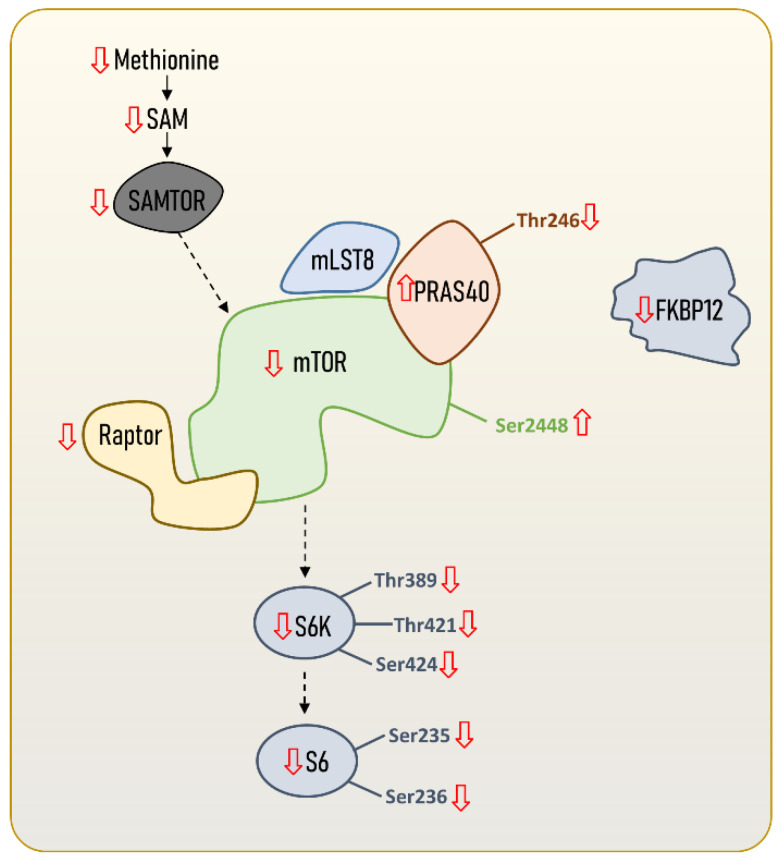
mTORC1 content and regulation in long-lived animal species, including humans. Steady-state levels of mTORC1 core elements and regulators are downregulated in long-lived species compared to short-lived ones. The content and regulation of other mTORC1 downstream proteins and transcript factors different from S6K and S6 are not included, since no data are available. However, scientific evidence suggests that there might also be specific regulations in downstream proteins or transcriptional factors targeting autophagy (e.g., ATG13, ULK1), mitochondrial biogenesis (e.g., PGC1α), lipids (e.g., SRBEP1, Lipin1, PP2A) and protein metabolism (e.g., 4E-BP1). Increased or reduced content of mTORC1 core elements and regulators can refer to either transcript content, gene expression, protein content, or metabolite content.

**Table 1 ijms-23-08747-t001:** Relationship between mTOR and maximum longevity (ML) in basal conditions: interspecies studies. Increased or reduced mTOR content can refer to transcript content, gene expression, protein content, protein phosphorylation, or activity after insulin stimulation. Letters refer to: maximum longevity (ML), in years; female (F); male (M); not determined (n.d.). Symbols refer to: * Studies in which mTORC1 itself expression, content or phosphorylation was evaluated; ^#^ Studies in which mTORC1 itself was not evaluated but its upstream or downstream effectors were.

Species (ML)	Sex	ML Range	mTORC1	Tissue	Phenotype	Ref.
Ames dwarf mice (5) vs. Wild type mice (3.5)	F	3.5 to 5	Reduced ^#^	Liver, skeletal muscle	Longevous	[22]
Naked mole rat (37) vs. Wild type mice (3.5)	F	3.5 to 30	Reduced ^#^	Liver	Longevous	[23]
8 mammals	M	3.5 to 46	Reduced *	Heart	Longevous	[14]
26 mammals	n.d.	2.1 to 37	Reduced *	Brain, heart, liver, kidney, lung and limb	Longevous	[25]
Human	M/F	89 to 102	Reduced *	Whole blood	Longevous	[26]

**Table 2 ijms-23-08747-t002:** Genetic interventions directed toward modifying the mTORC1 or mTOR signalling pathways. Genetic interventions refer to: heterozygous knockout (−/−), homozygous knockout (+/−), dominant-negative (d), hypomorphic (Δ/Δ), overexpression (gene), and human isoform (h). Increased or reduced mTOR content can refer to transcript content gene expression, protein content, protein phosphorylation, or activity after insulin stimulation. Longevity refers to percentage of maximum longevity (ML), in years. Letters refer to female (F); male (M); not determined (n.d.). Symbols refer to: * Studies in which expression, content, or phosphorylation of mTORC1 itself were evaluated; ^#^ Studies in which mTORC1 itself wasn’t evaluated, but its upstream or downstream effectors were. ^†^ Refers to percentage of change in average longevity (AL), in years.

Species	Sex	Genotype	mTORC1	Tissue	Phenotype	Longevity	Ref.
Worm	-	daf-15^−/−^	Reduced ^#^	Whole	n.d.	+28%	[31]
Worm	-	let-363^−/−^	Reduced *	Whole	n.d.	+250%	[32]
Fly	M	dTOR	Reduced ^#^	Whole	n.d.	+22–24%	[27]
Fly	M	dS6K	Reduced ^#^	Whole	n.d.	+22–24%	[27]
Fly	M	Tsc1/Tsc2	Reduced ^#^	Whole	n.d.	+12–14%	[27]
Mouse	F	hTsc1	Reduced ^#^	Heart, liver, kidney, skeletal muscle	Improved health	+12.3%	[28]
Mouse	F	*mtor* ^+/−^ *mlst8* ^+/−^	Reduced *	Liver	n.d.	+14.4	[33]
Mouse	F	mTOR^Δ/Δ^	Reduced *	Heart, liver, kidney, brain	Delayed ageing	+19% ^†^	[34]
Mouse	M	mTOR^Δ/Δ^	Reduced *	Heart, liver, kidney, brain	Delayed ageing	+22% ^†^	[34]
Mouse	F	s6k1^−/−^	Reduced ^#^	Liver, thymus, fibroblasts	Improved health	+19%	[29,30]
Mouse	n.d.	fkbp12^−/−^	Increased *	Hippocampus	Altered behaviour and neuronal function	n.d.	[35]

**Table 3 ijms-23-08747-t003:** Age-related changes in mTOR signalling. Increased or reduced mTOR content can refer to either transcript content, gene expression, protein content, protein phosphorylation or activity after insulin stimulation. Letters refer to: female (f); male (m). Symbols refer to: * Studies in which the expression, content or phosphorylation of mTORC1 itself were evaluated; ^#^ Studies in which mTORC1 itself wasn’t been evaluated, but its upstream or downstream effectors were.

Species	Sex	mTORC1	Tissue	Phenotype	Ref.
Mouse	F	Increased ^#^	Liver	Aged	[39]
Mouse	M	Increased ^#^	Liver	Aged	[39]
Mouse	M	Increased ^#^	Liver	Aged	[36]
Mouse	M	Increased *	Hematopoietic stem cells	Aged	[37]
Rat	M	Increased *	Skeletal muscle	Aged	[38]
Human	F/M	Increased *	Frontal cortex	Aged	[41]

## Data Availability

Not applicable.

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
