# Peer review of "mTOR Complex 1 Content and Regulation Is Adapted to Animal Longevity"

_ijms, 2022, doi:10.3390/ijms23158747_

Round 1

Reviewer 1 Report

The review is an extensive and very well researched summary of studies relevant to the role of mTTORC1 in regulating longevity as well as studies of inhibitors of mTORC1 as therapeutic intervention in age-associated diseases. The review is informative and interesting, but there are some difficulties in the language that occasionally obscure understanding. These are straightforward and I summarise them as follows:

1.     Line 54. The sentence beginning with “The latter…” describes the role of PRAS40, but there is no mention of the former, DEPTOR. There is no mention of DEPTOR in the review, which is surprising, considering that it is a direct inhibitor of mTORC1.

2.     Line 78. The clause “that define an mTORC1 active” makes no sense. Perhaps “There are specific modifications that are characteristic of active mTORC1.”

3.     Lines 79-80: These phosphorylations accompany mTORC1 activation, but is there any evidence that they activate mTORC1? If the sites are mutated, is mTORC1 inactive? This is an important issue that should be clarified.

4.     Line 82 “Less phosphorylation sites” should be “Fewer phosphorylation sites”

5.     Line 104 would be better as “Long-lived rodents [20,21], whales[23] and other mammals [22].”

6.     Table 1. Maybe there is a problem with reference 21. There are no measured longevities in the paper. In the cited reference, there is an unsourced statement about the longevity of the naked mole rat as up to 37 years. Apparently, the longevities quoted in the table are from another reference. These ML numbers should be given in parentheses next to the animals. They do not seem to be consistent. Wildtype mice are probably 5 years, yet this looks like the authors are saying that it is 30 years.

7.     Lines 118-120. The sentence is very awkward, and completely unintelligible. I am completely unclear on what the authors are trying to say.

8.     Line 125. According to the reference, all of these genes have decreased transcription levels associated with increased familial longevity. However, RAPTOR is not a negative regulator of mTOR. It is an essential component of mTORC1.

9.     Line 121. “In addition to…” would be better than “Besides…” The authors appear to like to begin sentences with “Besides”, yet in nearly every case, this makes it difficult understand what they have written. “In addition to…” might be a reasonable substitute often. Sometimes, it can completely be omitted.

10.  Line 142. “constituting elements itself” is awkward and unclear. I would suggest “components themselves”

11.  Line 143 “Besides” to “in addition”

12.  Line 145. “In the opposite…” change to “In contrast…”

13.  Line 153. Change “affect” to “operate”

14.  Lines 180-183. The sentence is impossible to understand. My best guess would be to change to “Interestingly, S6K1 activity and protein ubiquitination show a pattern of increase in middle age that is  similar to mTOR, while Akt phosphorylation upstream of mTORC1 and 4EBP1 phosphorylation downstream of mTORC1 increase continuously with age.”

15.  Line 195. Remove comma after “processes”. Delete “the” before ageing. Change “process leading” to “and lead”. Altogether the sentence should be: “Most of the mentioned processes are altered during ageing and lead to the development of age-related diseases.”

16.  Line 199. Remove “Besides”

17.  Line 199-200. Change “AD arising as” to “AD, making it”

18.  Line 202 “Besides…” to “In addition”

19.  Change “Besides from their neurological…” to “In addition to their potentially  beneficial neurological…”

20.  Line 205 effect” should be “effects”

21.  Line 205 Remove “intracellular”.

22.  Line 206 change “accordingly” to “Paradoxically…”. It is important to make clear that decreased muscle mass associated with increased mTOR activity in ageing would not be what would be expected, but it is what is observed with age associated sarcopenia.

23.  Line 207. Remove “are maintained and”

24.  End of line 207. “, as” change to “concomitant with”

25.  Line 208. Delete “occurs”

26.  Lines 260-261: “Besides, 1000d of moderate rapamycin treatment is…” Change to “Moderate rapamycin treatment for 1000d…”

27.  Line 373. Carrots, flowers and electrons in a synchrotron come in bunches, but I am not sure that evidence does. Change “A bunch of evidence” to “multiple lines of evidence”

28.  Table 6. The legend refers to “leucine restriction (LeuR); methionine restriction (MetR);”, however LeuR and MetR have not been used in the table.

Author Response

R: We would like to thank reviewers for their interest, helpful comments and suggestions. We have corrected the manuscript in order to address them, which we feel has clarified and improved it. New text parts (or changes) are highlighted throughout the manuscript in yellow. The detailed answers to the specific points are given below.

Reviewer 1

The review is an extensive and very well researched summary of studies relevant to the role of mTTORC1 in regulating longevity as well as studies of inhibitors of mTORC1 as therapeutic intervention in age-associated diseases. The review is informative and interesting, but there are some difficulties in the language that occasionally obscure understanding. These are straightforward and I summarise them as follows:

R: We appreciate the reviewer’s comments

  1. Line 54. The sentence beginning with “The latter…” describes the role of PRAS40, but there is no mention of the former, DEPTOR. There is no mention of DEPTOR in the review, which is surprising, considering that it is a direct inhibitor of mTORC1.

R: We understand the reviewer’s surprise. At first, we hadn’t included information regarding Deptor, as it is a common inhibitor of mTORC1 and mTORC2, and most of the longevity modulations had been attributed to mTORC1. However, we have included a sentence in line 54-55 (please see text highlighted in yellow). Although no information regarding the role of Deptor in regulating longevity, which is the main scope of the review, we have included a reference to its feasible role in regulating mTORC1 activity and the development of Alzheimer’s disease (please see text highlighted in yellow in lines 205-209).

  1. Line 78. The clause “that define an mTORC1 active” makes no sense. Perhaps “There are specific modifications that are characteristic of active mTORC1.”

R: We have corrected the sentence. Please see text highlighted in yellow in line 80-81.

  1. Lines 79-80: These phosphorylations accompany mTORC1 activation, but is there any evidence that they activate mTORC1? If the sites are mutated, is mTORC1 inactive? This is an important issue that should be clarified.

R: We agree with the reviewer. In fact, we had already tried to introduce this issue in lines 80-81, aimed to refer to the complexity of mTORC1 phosphorylation when elucidating its basal level of activation. However, we have included lines 84-87 (please see text highlighted in yellow) to remark the fact that phosphorylation at Ser2448 is not essential for mTORC1 activation.

  1. Line 82 “Less phosphorylation sites” should be “Fewer phosphorylation sites”

R: We have corrected the sentence. Please see text highlighted in yellow in line 87.

  1. Line 104 would be better as “Long-lived rodents [20,21], whales [23] and other mammals [22].”

R: We have corrected the sentence. Please see text highlighted in yellow in line 109.

  1. Table 1. Maybe there is a problem with reference 21. There are no measured longevities in the paper. In the cited reference, there is an unsourced statement about the longevity of the naked mole rat as up to 37 years. Apparently, the longevities quoted in the table are from another reference. These ML numbers should be given in parentheses next to the animals. They do not seem to be consistent. Wildtype mice are probably 5 years, yet this looks like the authors are saying that it is 30 years.

R: We agree with the misunderstanding. We have corrected the data in table 1 (see text highlighted in yellow)

  1. Lines 118-120. The sentence is very awkward, and completely unintelligible. I am completely unclear on what the authors are trying to say.

R: We have re-structured the paragraph. Please see text highlighted in yellow in lines 125-133.

  1. Line 125. According to the reference, all of these genes have decreased transcription levels associated with increased familial longevity. However, RAPTOR is not a negative regulator of mTOR. It is an essential component of mTORC1.

R: We have corrected this mistake. Please see text highlighted in yellow in line 127

  1. Line 121. “In addition to…” would be better than “Besides…” The authors appear to like to begin sentences with “Besides”, yet in nearly every case, this makes it difficult understand what they have written. “In addition to…” might be a reasonable substitute often. Sometimes, it can completely be omitted.

R: We have corrected the sentence. Please see text highlighted in yellow in line 129.

  1. Line 142. “constituting elements itself” is awkward and unclear. I would suggest “components themselves”

R: We have corrected the sentence. Please see text highlighted in yellow in line 147.

  1. Line 143 “Besides” to “in addition”

R: We have corrected the sentence. Please see text highlighted in yellow in line 148.

  1. Line 145. “In the opposite…” change to “In contrast…”

R: We have corrected the sentence. Please see text highlighted in yellow in line 150.

  1. Line 153. Change “affect” to “operate”

R: We have corrected the sentence. Please see text highlighted in yellow in line 158.

  1. Lines 180-183. The sentence is impossible to understand. My best guess would be to change to “Interestingly, S6K1 activity and protein ubiquitination show a pattern of increase in middle age that is  similar to mTOR, while Akt phosphorylation upstream of mTORC1 and 4EBP1 phosphorylation downstream of mTORC1 increase continuously with age.”

R: We have corrected the paragraph. Please see text highlighted in yellow in line 185-188.

  1. Line 195. Remove comma after “processes”. Delete “the” before ageing. Change “process leading” to “and lead”. Altogether the sentence should be: “Most of the mentioned processes are altered during ageing and lead to the development of age-related diseases.”

R: We have corrected the sentence. Please see text highlighted in yellow in line 200-201.

  1. Line 199. Remove “Besides”

R: We have corrected the sentence. Please see text highlighted in yellow in line 204.

  1. Line 199-200. Change “AD arising as” to “AD, making it”

R: We have corrected the sentence. Please see text highlighted in yellow in line 204.

  1. Line 202 “Besides…” to “In addition”

R: We have corrected the sentence. Please see text highlighted in yellow in line 214.

  1. Change “Besides from their neurological…” to “In addition to their potentially beneficial neurological…”

R: We have corrected the sentence. Please see text highlighted in yellow in line 214.

  1. Line 205 effect” should be “effects”

R: We have corrected the sentence. Please see text highlighted in yellow in line 214.

  1. Line 205 Remove “intracellular”.

R: We have corrected the sentence. Please see text highlighted in yellow in line 214.

  1. Line 206 change “accordingly” to “Paradoxically…”. It is important to make clear that decreased muscle mass associated with increased mTOR activity in ageing would not be what would be expected, but it is what is observed with age associated sarcopenia.

R: We have corrected the sentence. Please see text highlighted in yellow in line 216-217.

  1. Line 207. Remove “are maintained and”

R: We have corrected the sentence. Please see text highlighted in yellow in line 216.

  1. End of line 207. “, as” change to “concomitant with”

R: We have corrected the sentence. Please see text highlighted in yellow in line 216.

  1. Line 208. Delete “occurs”

R: We have corrected the sentence. Please see text highlighted in yellow in line 217.

  1. Lines 260-261: “Besides, 1000d of moderate rapamycin treatment is…” Change to “Moderate rapamycin treatment for 1000d…”

R: We have corrected the sentence. Please see text highlighted in yellow in line 269-270.

  1. Line 373. Carrots, flowers and electrons in a synchrotron come in bunches, but I am not sure that evidence does. Change “A bunch of evidence” to “multiple lines of evidence”

R: We have corrected the sentence. Please see text highlighted in yellow in line 382.

  1. Table 6. The legend refers to “leucine restriction (LeuR); methionine restriction (MetR);”, however LeuR and MetR have not been used in the table.

R: We have removed these abbreviations from the table 6 description. Please see text highlighted in yellow in line 414-416.

Reviewer 2 Report

A timely review article by Dr. Mota-Martorell and Pamplona elaborates on the role of mTORC1 and its role in animal life span. This is a well-written review article, though a few things need to be addressed before it is ready for acceptance. They are as follows:

1. It has been shown that AMPK regulates mTOR activity in a feedback loop manner mediated by Phosphatidic acid and this opens up the usage of AMPK activators like metformin along with rapamycin as therapeutic agents (PMID: 26323019 and PMID: 22017684). Authors should discuss this aspect by referring to the relevant articles in the context of the current review manuscript.

2. It has also been discussed how mTOR plays a significant role in metabolic checkpoint regulation (PMID: 26682255 and PMID: 30131808). Authors should add a few lines on these points by mentioning the relevant references.  

3. It has also been discussed in some aspects dosage of rapamycin is a determinant factor (PMID: 26916116 and PMID: 25269671). This point must be discussed in a few lines.

4. Fig 2 should be more elaborative by adding other signaling pathways

related to the context of this manuscript.

5. Authors should add a table by mentioning the relevant clinical trials going on related to the theme of tithis

manuscript.

Author Response

We would like to thank reviewers for their interest, helpful comments and suggestions. We have corrected the manuscript in order to address them, which we feel has clarified and improved it. New text parts (or changes) are highlighted throughout the manuscript in yellow. The detailed answers to the specific points are given below.

Reviewer 2

A timely review article by Dr. Mota-Martorell and Pamplona elaborates on the role of mTORC1 and its role in animal life span. This is a well-written review article, though a few things need to be addressed before it is ready for acceptance. They are as follows:

R: We appreciate the reviewer’s comments

  1. It has been shown that AMPK regulates mTOR activity in a feedback loop manner mediated by Phosphatidic acid and this opens up the usage of AMPK activators like metformin along with rapamycin as therapeutic agents (PMID: 26323019 and PMID: 22017684). Authors should discuss this aspect by referring to the relevant articles in the context of the current review manuscript.

R: We appreciate the reviewer’s comment, but we would like to keep our attention in mTOR. We haven’t mentioned upstream signalling pathways that regulate mTOR, as we aimed to describe the basal status of mTOR itself, and how its core elements are regulated under longevity evolution. Besides, these upstream signalling pathways respond to changing conditions, such as nutrient availability, and not to basal and physiological status of mTOR.

  1. It has also been discussed how mTOR plays a significant role in metabolic checkpoint regulation (PMID: 26682255 and PMID: 30131808). Authors should add a few lines on these points by mentioning the relevant references.  

R: We have included this issue. Please see text highlighted in yellow in lines 400-405.

  1. It has also been discussed in some aspects dosage of rapamycin is a determinant factor (PMID: 26916116 and PMID: 25269671). This point must be discussed in a few lines.

R: We have already discussed this issue in lines 236-241, 263-273 and 274-280. As we believe that we have deeply described the need to properly adjust rapamycin dosage and pattern of administration, we have included a sentence referring to the two mentioned papers (please see text highlighted in yellow in line 280).

  1. Fig 2 should be more elaborative by adding other signaling pathways related to the context of this manuscript.

R: As for comment 1, we appreciate reviewer’s comment, but we would like to keep our attention in the regulation of mTORC1 itself. In fact, we had already included the information of several basal downstream effectors of mTOR (please see text highlighted in yellow in lines 139-141.

  1. Authors should add a table by mentioning the relevant clinical trials going on related to the theme of this manuscript.

R: We appreciate reviewer’s comments. Few clinical trials aimed to delay the ageing process targeting to mTOR have been performed. In fact, to the best our knowledge, there is only one clinical trial ongoing aimed to evaluate the effect of rapamycin on physiological ageing. Accordingly, we have included a new section in our review (please see text highlighted in yellow in lines 424-445). As we stated, little information is known and we opted for including a new section rather than generating a new table (our manuscript already includes up to 6 tables).

Round 2

Reviewer 2 Report

All concerns have been addressed, ready for acceptance. 

Author Response

We appreciate the reviewer's comment